# Passage-specific Prompt Tuning for Passage Reranking in Question Answering with Large Language Models

Xuyang Wu*
Santa Clara University
Santa Clara, USA
xwu5@scu.edu

Zhiyuan Peng*
Santa Clara University
Santa Clara, USA
zpeng@scu.edu

Krishna Sravanthi Rajanala Sai
Walmart Global Tech
Sunnyvale, USA
sravanthi.rajanala@walmart.com

Hsin-Tai Wu
Docomo Innovations
Sunnyvale, USA
hwu@docomoinnovations.com

Yi Fang
Santa Clara University
Santa Clara, USA
yfang@scu.edu

## ABSTRACT

Effective passage retrieval and reranking methods have been widely utilized to identify suitable candidates in open-domain question answering tasks, recent studies have resorted to LLMs for reranking the retrieved passages by the log-likelihood of the question conditioned on each passage. Although these methods have demonstrated promising results, the performance is notably sensitive to the human-written prompt (or hard prompt), and fine-tuning LLMs can be computationally intensive and time-consuming. Furthermore, this approach limits the leverage of question-passage relevance pairs and passage-specific knowledge to enhance the ranking capabilities of LLMs. In this paper, we propose passage-specific prompt tuning for reranking in open-domain question answering (PSPT[1]): a parameter-efficient method that fine-tunes learnable passage-specific soft prompts, incorporating passage-specific knowledge from a limited set of question-passage relevance pairs. The method involves ranking retrieved passages based on the log-likelihood of the model generating the question conditioned on each passage and the learned soft prompt. We conducted extensive experiments utilizing the Llama-2-chat-7B model across three publicly available open-domain question answering datasets and the results demonstrate the effectiveness of the proposed approach.

## 1 INTRODUCTION

Open-domain question answering (QA) involves to answer questions from a vast collection of passages [41]. The existing works [5, 12, 44] have demonstrated that efficiently retrieving a small subset of passages, which contain the answer to the question, is a crucial part of enhancing the QA task. Typically, relevant passages can be retrieved using keyword matching methods such as TF-IDF or BM25 [31], or through dense latent representations [5]. The results can be refined further by reranking the top-$k$ retrieved passages to ensure accuracy.

---

*Both authors contributed equally to this research.
[1]Our code can be found at https://github.com/elviswxy/Gen-IR_PSPT.

Generative text reranker (GTR) resort to the model's generation ability to rerank the retrieved passages. By leveraging the powerful generation capabilities of Large Language Models (LLMs), GTR has demonstrated cutting-edge reranking performances, even directly output the permutation of input documents (or passages) based on their relevances to the given query (or question) [35]. Current GTR methods either fine-tune the whole large language model on question-passage relevance pairs [6, 19, 48] or rely on prompt engineering to craft good prompts for producing desirable output [20, 26, 32, 35]. While it is possible to fine-tune the whole LLMs like T0-3B [34], it becomes prohibitively computationally intensive and time-consuming on larger and advanced LLMs such as Llama-2 [19, 40]. On the other hand, the prompt engineering approach to LLMs saves cost but the results are highly sensitive to both the quality of human-written prompt (hard prompt) and the generation ability of LLMs. Moreover, hard prompting cannot benefit from the available question-passage relevance pairs of passage-specific knowledge.

In this paper, we propose passage-specific prompt tuning for passage reranking in open-domain question answering (PSPT), a parameter-efficient method that fine-tunes learnable passage-specific soft prompts on a limited set of question-passage relevance pairs. Specifically, our method involves integrating a soft prompt with a passage-specific embedding layer to form a new learnable prompt module, then concatenated with the embeddings of the raw passage to serve as new input for the LLMs. We maximize the log-likelihood of a query conditioned on the relevant or positive passage and utilize a hinge loss to punish the model when the log-likelihood of a query conditioned on the positive passage is smaller than that conditioned on the negative passage. Only a small proportion of learnable parameters $\theta$ are updated during the training while LLM's parameters $\Phi$ are fixed. While recent research has successfully applied LLMs such as ChatGPT to reranking [35], most of the existing work has been built on proprietary models hidden behind opaque API endpoints, which may produce non-reproducible or non-deterministic experimental results [26]. Instead, our work is based on a popular open-source large language model (LLM): Llama-2 [40]. Our main contributions can be summarized as follows:

- To the best of our knowledge, this is the first work to enhance soft prompt tuning with passage-specific knowledge for passage reranking in QA tasks with LLMs.

- Our parameter-efficient tuning approach based on an open-source LLM substantially enhances reranking performance in QA with minimal parameter increments while preserving reproducibility.
- A comprehensive set of experiments is conducted on three widely used open-domain QA datasets. The results demonstrate that integrating a soft prompt module with a passage-specific embedding layer significantly enhances reranking performance beyond that of baseline retrievers and recently published models based on LLMs.

## 2 RELATED WORK

### 2.1 Passage Retrieval and Reranking

The QA systems utilize passage retrieval and reranking to select candidates. The unsupervised retrievers, like BM25 [31], MSS [33], and Contriever [9], and supervised retrievers, such as DPR [12], and MSS-DPR [33], retrieve a subset of candidate passages, which are then re-ranked in subsequent stages. BM25 measures the relevant scores between questions and passages by exact term matching. MSS is a dense retriever that undergoes joint training with both retriever and reader components. Contriever utilizes a contrastive learning framework for information retrieval (IR) pre-training, exhibits competitive performance relative to BM25. DPR is a dense passage retriever that uses the bi-encoder architecture, trained on question-passage relevance pairs. MSS-DPR presents a notable enhancement in DPR's efficacy through pre-training the dense retriever with the MSS method and applying DPR-style supervised fine-tuning. Besides these retrievers, recently, some new supervised methods are proposed like ColBERT [13], SPLADE [7] and SparseEmbed [14]. In this study we rerank the top-$k$ passages retrieved by BM25, MSS, Contriever, DPR and MSS-DPR to show its effectiveness across different retrieval paradigms.

GTR converts a reranking task into a generation task to utilize LLMs. Such as, query generation [2, 6, 32, 49], relevance generation [18, 47, 48], and permutation generation [20, 26, 27, 35, 37, 42]. The query generation method computes the relevance score between a query and a document by the log-likelihood of LLMs to generate the query based on that document. UPR [32] calculates the query generation log-likelihood based on T0-3B [34] while dos Santos et al. [6] fine-tune GPT-2 [28] and BART [17] with unlikelihood loss and pairwise loss respectively and then computed the query generation log-likelihood. The relevance generation method [19, 35, 48] adopts the logit on certain word, like "yes", "no" or "\s", as the relevance score. The permutation generation methods prompts LLMs to directly output the ordered documents ranked by relevance. RankVicuna [26] to output the document order directly. Similarly, LRL [20] prompts GPT-3 [1] for ranking input documents. Our method is different from the existing GTR methods above in that we learn an efficient passage-specific prompt module on limited question-passage relevance pairs to enhance LLM's strong generation ability and guide the LLMs in the passage reranking of QA task.

### 2.2 Prompt Tuning

Prompt Tuning is a parameter-efficient technique that adapts pretrained LLMs for specific tasks by adjusting the prompt module,

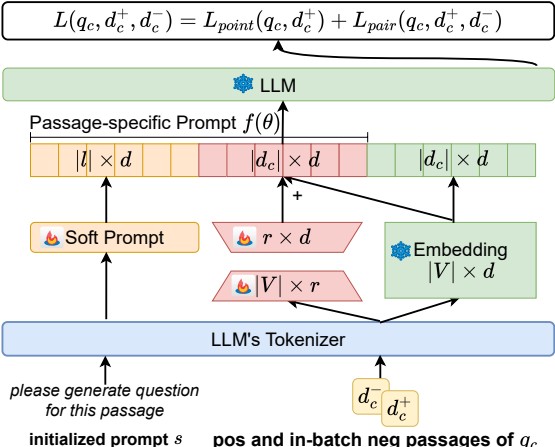

$$L(q_c, d_c^+, d_c^-) = L_{point}(q_c, d_c^+) + L_{pair}(q_c, d_c^+, d_c^-)$$

**Figure 1: The architecture of PSPT. The original LLM parameters $\Phi$ (green blocks) are frozen during training, with only $\theta$ parameters (red and yellow blocks) updated.**

rather than fine-tuning the entire model [16]. Some notable techniques include Prefix-Tuning [36], which prepends learnable embeddings to input tokens, "gisting" [22] which compresses prompts using a meta-learning approach, and task-specific prompt embeddings [16, 29] which incorporate task-specific information. SPTAR [25] optimizes a task-specific soft prompt to guide LLMs in tagging documents with weak queries, generating pairs that improve task-specific retriever performance. DCCP [46] leverages both prompt information and context to boost performance. ATTEMPT [43] transfers knowledge across tasks using a mixture of soft prompts. For Information Retrieval tasks, DPTDR [38] employs dual encoders and learnable soft prompts to enhance retrieval models. T5 [24], a text-to-text language model, models relevance in ranking tasks using document-query pairs and "true/false" label. Our method extends the soft prompt tuning approach from [16] by learning a passage-specific soft prompt, thereby enhancing the reranking capabilities of LLMs in QA tasks.

## 3 PASSAGE-SPECIFIC PROMPT TUNING

PSPT only fine-tunes a small number of parameters $\theta$ while keeping LLM's original parameters $\Phi$ fixed, learning a soft prompt and a set of embeddings in the training process. Subsequently, it reranks passages based on the log-likelihood of the question, conditioned on each retrieved passage along with the learned prompt. This method sets itself apart from the prompt tuning or soft prompt technique described in [36]. While the method in [36] employs a soft prompt that is consistent across all passages within the same dataset and varies only across different tasks or datasets, PSPT enriches this approach by incorporating passage-specific knowledge into the learned prompt, thereby boosting adaptability across a diverse range of passages. Consequently, the learnable prompt in PSPT dynamically adjusts not just to different tasks but also to individual passages.

Figure 1 illustrates the architecture of PSPT. The yellow blocks in Figure 1 are to learn a task-specific soft prompt $e_1$, and we follow

[36] to initialize the soft prompt by extracting the LLM's original embeddings of prompt $s$ "*please generate question for this passage*" which is repeated until the length of $s$ equals pre-defined soft prompt length $l_s$. Suppose the dimensionality of the embeddings is $dim$; then the learned $e_1$ has shape $|l_s| \times dim$. Apart from the task-specific soft prompt, PSPT employs the red blocks in Figure 1 to learn a prompt $e_2$ with shape $|d_i| \times dim$ for a passage $d_i$. Instead of learning a new embedding layer with $V \times d$ weights for passages, inspired by LoRA [8], we decompose $V \times dim$ by the product of two low-rank matrices $|V| \times r$ and $r \times dim$ which are initialized by random Gaussian and zero respectively to reduce the number of learnable parameters. For a passage $d_i$, we first look up its embeddings $|d_i| \times r$ in learnable embedding layer $|V| \times r$ and then product it with $r \times dim$ to get $e_3$ with shape $d_i \times dim$. Finally, we obtain $e_2 = e_3 * (\alpha/r) + e_4$ where $e_4$ represents the embeddings of $d_i$ encoded by LLM's original embedding layer and $\alpha$ helps to reduce the need to re-tune hyper-parameters when we vary $r$ [45]. We concatenate $e_1$, $e_2$ and $e_4$ as the input of LLM to compute the log-likelihood of question $q$ conditioned on passage $d$ and passage-specific prompt $f_\theta(s, d)$ which is defined as:

$$I_{\theta,\Phi}(q|s,d) = \sum_{l=1}^{|q|} \log P_{\theta,\Phi}\left(q_l \mid q_{<l}, s, d\right) \quad (1)$$

where $P_{\theta,\Phi}\left(q_l \mid q_{<l}, s, d\right)$ represents the possibility of predicting the current token by looking at previous tokens. For a dataset of questions, each of which has some positive and negative passages, we follow [12] to sample one positive passage and one negative passage for each question and apply in-batch negative strategy to generate more negative passages. We can assume the training data is a collection of instances $\left\langle q_i, d_i^+, d_i^- \right\rangle_{i=1}^{i=N}$.

Pointwise loss is applied to constrain the model to output a ground-truth-like question based on the input positive or relevant passage, which is defined on one instance $\left\langle q_i, d_i^+, d_i^- \right\rangle$ as:

$$L_{point}(q_i, d_i^+) = -I_{\theta,\Phi}(q_i|s, d_i^+) \quad (2)$$

To improve the model's ranking ability, inspired by hinge loss, on one instance $\left\langle q_i, d_i^+, d_i^- \right\rangle$, we also apply a pairwise loss $L_{pair}$ which is defined as:

$$L_{pair}(q_i, d_i^+, d_i^-) = \max\left\{0, I_{\theta,\Phi}(q_i|d_i^-, s) - I_{\theta,\Phi}(q_i|d_i^+, s)\right\} \quad (3)$$

Our final loss $L$ directly combines the pointwise and pairwise losses:

$$L(q_i, d_i^+, d_i^-) = L_{point}(q_i, d_i^+) + L_{pair}(q_i, d_i^+, d_i^-) \quad (4)$$

During the inferencing process, the PSPT model reranks the top-$k$ passages, denoted as $z_1, z_2, ...z_k$, which are retrieved by the retriever $R$. The relevance score for this reranking is based on the log-likelihood of the generated question $q$ conditioned on each passage $z_j$ along with the leaned passage-specific prompt. This is expressed as $I_{\theta^*,\Phi}(q|s, z_j)$, where $s$ represents the initialized prompt and $\theta^*$ denotes the learned optimal parameters after training phase.

## 4 EXPERIMENTS

### 4.1 Datasets, Baselines and Evaluation Metrics

Following the existing work of DPR [12] and aiming for fair comparisons, our study utilized the QA datasets presented in Appendix A.6.

| Retriever | NQ | | SQuAD | | TriviaQA | |
|---|---|---|---|---|---|---|
| | R@10 | H@10 | R@10 | H@10 | R@10 | H@10 |
| Unsupervised Retrievers | | | | | | |
| BM25 | 22.01 | 49.94 | 26.33 | 49.67 | 22.85 | 62.77 |
| +UPR | 32.31 | 59.45 | 42.33 | 64.78 | 36.17 | 71.80 |
| +UPR-Inst | 31.75 | 58.86 | 42.04 | 64.65 | 36.37 | 71.59 |
| +PSPT | †‡**36.89** | †‡**62.24** | †‡**46.04** | †‡**66.76** | †‡**42.63** | †‡**73.71** |
| MSS | 19.19 | 51.27 | 20.28 | 42.51 | 19.97 | 60.52 |
| +UPR | 33.71 | 63.91 | 38.22 | 60.14 | 37.44 | 72.29 |
| +UPR-Inst | 33.35 | 63.19 | 37.77 | 59.76 | 38.01 | 72.70 |
| +PSPT | †‡**37.95** | †‡**66.45** | †‡**41.80** | †‡**62.20** | †‡**44.30** | †‡**74.68** |
| Contriever | 22.31 | 58.73 | 26.06 | 54.65 | 20.27 | 68.00 |
| +UPR | 32.38 | 67.12 | 41.25 | 69.06 | 32.58 | 75.86 |
| +UPR-Inst | 31.27 | 66.07 | 40.75 | 68.74 | 32.90 | 75.74 |
| +PSPT | †‡**37.45** | †‡**70.42** | †‡**46.09** | †‡**72.16** | †‡**39.46** | †‡**78.03** |
| Supervised Retrievers | | | | | | |
| DPR | 38.74 | 74.54 | 25.68 | 51.42 | 27.93 | 76.50 |
| +UPR | 41.73 | 75.60 | 41.57 | 66.01 | 36.83 | 80.28 |
| +UPR-Inst | 40.28 | 74.46 | 41.51 | 65.94 | 36.88 | 80.19 |
| +PSPT | †‡**45.73** | †‡**77.84** | †‡**45.27** | †‡**68.45** | †‡**42.53** | †‡**81.67** |
| MSS-DPR | 37.47 | 77.48 | 33.25 | 65.85 | 25.54 | 79.15 |
| +UPR | 38.79 | 76.81 | 46.96 | 77.10 | 31.33 | 81.56 |
| +UPR-Inst | 37.16 | 75.35 | 46.88 | 76.93 | 31.44 | 81.68 |
| +PSPT | †‡**43.02** | †‡**79.09** | †‡**51.23** | †‡**79.36** | †‡**36.24** | †‡**82.83** |

**Table 1: The symbols † and ‡ indicate statistically significant improvements over basic retrievers and the UPR approach, respectively, determined by t-test with p-values < 0.05.**

We selected the Unsupervised Passage Retrieval (UPR) approach as a competitive baseline model. UPR utilizes a pre-trained language model to estimate the probability of an input question conditioned on a retrieved passage. Specialized, we replace the pre-trained language model in UPR with Llama-2-chat-7B to examine its capabilities and performance, ensuring a fair comparison. Furthermore, by leveraging the instruction tuning strategy, we use a high-quality question-passage pair to guide the generation process in UPR, aiming to enhance its performance. We name this baseline UPR-Inst. In the Appendix A.3, we provide detailed descriptions of the instruct prompt formats used by the baseline models.

In our work, we utilized the Llama-2-Chat model with 7 billion parameters. We employed the top-$k$ Recall (R@$k$) and Hit Rate (H@$k$) to evaluate the reranking performance.

### 4.2 Implementation Details

For the baseline models, we used the base configuration as specified in each respective paper. We implemented PSPT based on the publicly available prompt tuning package PEFT [21]. We choose hard prompt initialization $s$ as "please generate question for this passage" with pre-defined soft prompt length $l_s = 50$. For hyper-parameters, $r = 1$ and $\alpha = 16$ are selected for learning passage-specific embedding $e_2$ in Figure 1. We fine-tuned the PSPT model on Nvidia A100 GPUs, using the bfloat16 [11] data type, across different training sample sizes ranging from 320 to 1280 for each dataset. The training involved a batch size of 4 and an in-batch negative sampling. The learning rates for yellow blocks and red blocks in Figure 1 are set at 3e-2 and 3e-5, respectively, with linear decay. We trained PSPT 20

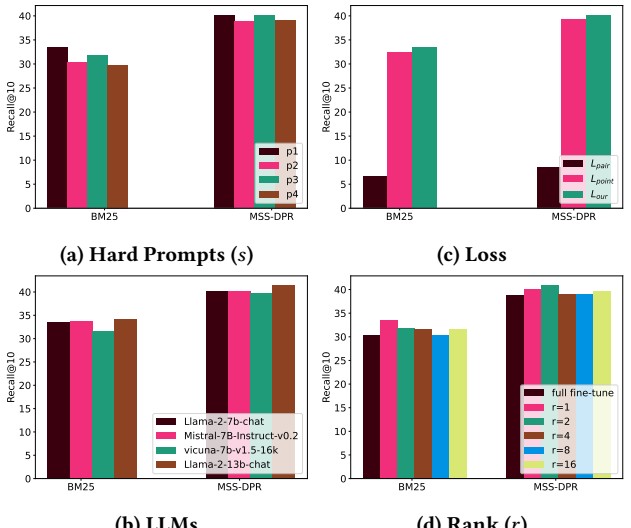

(a) Hard Prompts ($s$)

(c) Loss

(b) LLMs

(d) Rank ($r$)

**Figure 2: Performance analysis of key components in PSPT on the sampled NQ Dataset.**

| Method | Trainable | BM25 | | MSS-DPR | |
|---|---|---|---|---|---|
| | | R@10 | H@10 | R@10 | H@10 |
| Retriever Only | – | 16.58 | 43.67 | 35.91 | 74.67 |
| Hard Prompt Only (HP) | 0.0000% | 28.87 | 55.31 | 36.41 | 75.00 |
| Soft Prompt Only (SP) | 0.0007% | 29.95 | 56.33 | 38.40 | 76.00 |
| HP + passage-specific (FT) | 1.9080% | 29.34 | 55.67 | 36.81 | 73.67 |
| HP + passage-specific (LoRA) | 0.0005% | 29.52 | 56.00 | 37.92 | 75.00 |
| SP + passage-specific (FT) | 1.9110% | 30.35 | 55.33 | 38.86 | 75.67 |
| SP + passage-specific (LoRA) | 0.0012% | **33.44** | **59.33** | **40.03** | **76.33** |

**Table 2: Comparison of PSPT modules on the sampled NQ Dataset. Trainable parameters relative to Llama-2's total parameters are presented. FT and LoRA denote fully fine-tuning and LoRA-based tuning of the passage-specific module, respectively.**

(4) Using solely $L_{point}$ or $L_{pair}$ does not yield optimal results, as $L_{pair}$ does not aim to optimize effective generation capabilities, and $L_{point}$ is primarily optimized based only on positive sample data. Combining both losses enables the model to understand ranking orders and boosts generation effectiveness, which further improves model performance (in Figure 2c).

### 4.4 Ablation Study

We performed a detailed comparative analysis of various modules within PSPT to evaluate their efficiency and effectiveness. As shown in Table 2, for each experiment, we given the proportion of trainable parameters relative to Llama-2-chat-7B's total parameters to illustrate the fine-tuning process's efficiency. Our findings can be summary as: (1) All methods are capable of enhancing ranking performance of MSS-DPR and BM25 retrieval methods; (2) Converting hard prompts into trainable soft prompts enhances the performance; (3) Updates the embedding layer parameters of passage-specific module using the LoRA-based technique is more effective than fully fine-tuning of all parameters, regardless of the type of prompts used. However, this approach was slightly less effective than experiments using only soft prompts as more trainable parameters are needed. (4) Integrating the soft prompt module with the LoRA-based embedding layer configuration obtain the best performance.

### 5 CONCLUSION AND FUTURE WORK

In this paper, we present a parameter-efficient passage-specific prompt tuning approach to enhance LLMs for passage reranking in QA. Through extensive experimentation across three datasets, we demonstrate the effectiveness of our proposed PSPT. For future research, we will conduct experiments on more and larger datasets like MS MARCO[23], TREC2019[4], TREC2020[3] and BEIR[39] to thoroughly assess PSPT's generalizability in domains other than QA, and compare with more relevant state-of-the-art baseline models. Furthermore, since PSPT operates through a plug-in mechanism, adopting a learnable prompt module to enhance the probability of a frozen LLM generating true queries, we can combine our learnable prompt module with other methods, like LoRA [8] or more advanced ranking LLMs, like RankLLaMA [19], to further enhance ranking capabilities by fine-tuning the entire LLM.

epochs with early stopping. We have displayed additional results of hyper-parameter tuning in Appendix A.

### 4.3 Experimental Results

Table 1 presents a detailed evaluation of our proposed PSPT model's performance, showing that PSPT consistently surpasses both basic retrievers and baseline models. Essentially, our PSPT model achieves notable improvements across both unsupervised and supervised retrievers from three distinct datasets. This illustrates the powerful adaptability of our proposed model, capable of accommodating various potential datasets and retrieval environments. On the other hand, comparing the experimental results of unsupervised and supervised retrievers, firstly, our model can significantly enhance the reranking performance on the basis of the results from unsupervised retrievers. When the results from supervised retrievers are already good, the UPR-based model does not achieve consistent improvements in reranking performance, like on dataset NQ, UPR's H@10 is lower than that of MSS-DPR. In contrast, our model shows stable performance improvements across all datasets.

Additional findings are presented in Figure 2: (1) The PSPT module can continue to improve along with the enhancement of LLMs, such as the Llama-13B model with more parameters or the more powerful Mistral-7B model (in Figure 2b); (2) The soft prompt module demonstrates sensitivity to the initialization of hard prompts (in Appendix A.2), we selected only the best-performing initialization in our experiments (in Figure 2a). Furthermore, increasing the virtual prompt token length appropriately can provide additional space for the soft prompt module to adapt to new tasks during training (in Appendix A.1); (3) The passage-specific module effectiveness is sensitive to the extent of parameter changes. The LoRA-based technique offer a better alternative to full fine-tuning, as increase the rank $r$ leads to slight worse performance. In addition, experiments using the LoRA-based approach consistently outperform the fully fine-tuning of the passage-specific module (in Figure 2d);

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

# A  APPENDIX

## A.1  Pre-defined Soft Prompt Length

| Soft Prompt Length ($l_s$) | BM25 | | MSS-DPR | |
|---|---|---|---|---|
| | R@10 | H@10 | R@10 | H@10 |
| Retriever Only | 16.58 | 43.67 | 35.91 | 74.67 |
| $l_s = 20$ | 30.10 | 57.33 | 36.94 | 72.00 |
| $l_s = 30$ | 31.06 | 57.33 | 39.32 | 74.67 |
| $l_s = 40$ | 30.34 | 55.00 | 39.08 | 75.00 |
| $l_s = 50$ | **33.44** | **59.33** | **40.03** | **76.33** |
| $l_s = 60$ | 32.27 | 57.33 | 39.20 | 75.33 |
| $l_s = 80$ | 31.26 | 57.00 | 39.38 | 75.33 |
| $l_s = 100$ | 29.75 | 56.00 | 39.95 | 76.00 |

**Table 3: Performance comparison of PSPT modules on BM25, MSS-DPR, evaluated on the sampled NQ Dataset using Recall (R@10) and Hit Rate (H@10), highlighting the best results in bold. Each experiment demonstrates the impact of different pre-defined soft prompt virtual lengths on the experimental results, and we selected the best performance with $l_s = 50$.**

## A.2  Initialization of Hard Prompts

| Name | Content |
|---|---|
| p1 | Please generate question for this passage. |
| p2 | Generate a question based on the content of this passage. |
| p3 | Kindly craft a question based on the content provided in this passage. |
| p4 | Craft questions based on the provided passage. |

**Table 4: Different initialization of hard prompts utilized in PSPT.**

## A.3  Instruction Prompt of Baselines

For the UPR model, we used a fixed hard prompt as the instruction prompt. For the UPR-inst model, based on the hard prompt of the UPR model, we have added high-quality question-passage pairs to guide the generation process of the UPR model. We select these high-quality question-passage pairs for each dataset based on the top 3 BM25 results, and these instructive question-passage pairs will not appear in the training and testing data. The data formatted as follows:

| Baselines | Prompt Format |
|---|---|
| UPR | *Please generate question for this passage:*
*Passage: [Passage]*
*Question:* |
| UPR-inst | *Please generate question for this passage based on the example:*
*Example:*
*Passage: [Passage]*
*Question: [Question]*
*Passage: [document]*
*Question:* |

**Table 5: Instruction Prompt of Baselines.**

## A.4  In-batch Hard Negative Sample Size

| In-batch Negative Sampling | BM25 | |
|---|---|---|
| | R@10 | H@10 |
| Retriever Only | 16.58 | 43.67 |
| 2 | 31.37 | 58.33 |
| 4 | **33.44** | **59.33** |
| 8 | 28.54 | 53.33 |
| 16 | 26.4 | 54.33 |
| 32 | 20.9 | 47.67 |

**Table 6: Performance comparison of PSPT modules on BM25, evaluated on the sampled NQ Dataset using Recall (R@10) and Hit Rate (H@10), highlighting the best results in bold. Each experiment demonstrates the impact of different in-batch hard negative sample sizes on the experimental results. We chose the best performance with an in-batch hard negative sample size of 4.**

## A.5  Different Training Samples

| Training Sample Size | BM25 | | MSS-DPR | |
|---|---|---|---|---|
| | R@10 | H@10 | R@10 | H@10 |
| Retriever Only | 16.58 | 43.67 | 35.91 | 74.67 |
| 80 | 29.11 | 57.00 | 36.15 | 72.67 |
| 160 | 29.70 | 55.33 | 37.61 | 72.00 |
| 320 | **33.44** | **59.33** | 40.03 | 76.33 |
| 640 | 32.30 | 58.00 | 39.43 | 75.00 |
| 1280 | 30.92 | 56.33 | **40.10** | **76.67** |

**Table 7: Performance comparison of PSPT modules on BM25, MSS-DPR, evaluated on the sampled NQ Dataset using Recall (R@10) and Hit Rate (H@10), highlighting the best results in bold. Each experiment demonstrates the impact of different training sample sizes on the experimental results. Based on the model's training efficiency and effectiveness, we selected a training sample size of 320.**

## A.6  Data Description

| Dataset | Train | Ret.Train | Eval | Test |
|---|---|---|---|---|
| Natural Questions [15] | 79,168 | 58,880 | 8,757 | 3,610 |
| TriviaQA [10] | 78,785 | 60,413 | 8,837 | 11,313 |
| SQuAD [30] | 78,713 | 70,096 | 8,886 | 10,570 |

**Table 8: The statistics of the datasets. The column Ret.Train refer to the actual questions used for training supervised retrievers after filtering in the dataset.**