# OpenReview forum: "Passage-specific Prompt Tuning for Passage Reranking in Question Answering with Large Language Models"
_ACM.org/SIGIR/2024/Workshop/Gen-IR — Gen-IR_SIGIR24_

### Official Review · Reviewer_RVKX · 2024-05-23
**The paper provides numerous insights, and the evaluation results are well-aligned with the motivation of the methods.**

**Rating:** 2
**Confidence:** 4

**Review:**

- Quality: The quality is good in various aspects. The experiments in the paper are clear and well-organized; the methods and evaluation results are well-aligned;

- Clarity: Good. Only a few trivial settings are not reported. (e.g., top-K retrieved passages, negative documents). The other parts are clear.

- Originality: Good. The work explores the efficient fine-tuning with LLM, which is relatively new compared to popular previous works using prompts. (hard or in-context examples)

- Significance: the work empirically validate new alternatives of using LLMs and provide comprehensive insights in the ablation.

- Strength and comments
    * The paper is well-written and easy to understand.
    * The paper sheds lights on different way of utilizing LLM, providing many useful insights; the motivation of soft-prompt tuning and LoRA is well-aligned to the propositions (training efficient).
    * The proposed methods are validated and show the effective empirical results on open-retrieval question answering tasks.
    * The re-ranking experiments are validated on different first-stage retrieval, showing the robustness of proposed methods.
    * The experimental designs are rigorous and well-defined, supporting the results concretely.
    * The ablation studies provide several insights that can facilitate future works on IR with LLMs, such as data augmentation, pseudo-labeling, learning-to-rank with LLMs
    * It’s very interesting to see how this method will perform on other more benchmarks.

- Weakness, comments and suggestions
    * [Sec. 3] The pairwise ranking objective may rely heavily on negative document; however, the negatives are less discussed. —> It’s good to have some comparison between P(q|d+) and P(q|d-)
    * [Sec. 4.2/Appendix A.4] The description of negative is not clear enough. —> Naive in-batch negative more like a random negative? Or did the authors use negative mining strategies to obtain harder negative document?
    * [Sec. 4.2] the setting of LoRA’s r=1 is not intuitive or needs more elaboration. —> Does r=1 indicate that additional passage-specific signals from different passages are similar (n \times d)? It would be helpful to have some analysis of e2 and e3 (e.g., re-ranking effectiveness or some qualitative analysis).
    * [Sec. 4.3] Although the baselines have already demonstrated effectiveness, reporting on other re-ranking systems may help readers better understand the training efficiency.

---

### Official Review · Reviewer_P7c6 · 2024-05-25
**Review (Passage-specific Prompt Tuning for Passage Reranking in Question Answering with Large Language Models)**

**Rating:** -1
**Confidence:** 5

**Review:**

This paper inspired by the success of LORAs in LLMs, introduces a novel approach called Passage-specific Prompt Tuning (PSPT) for improving passage reranking in open-domain question-answering ranking tasks.
By incorporating learnable soft prompts, the method enhances the reranking capabilities of models with minimal parameter tuning (sometimes ~ 0.001%).
Extensive experiments across three datasets demonstrate PSPT's effectiveness, showing improvements over baseline retrievers.

## Strengths
1) Very parameter efficient! Getting a good bump with just training ~0.001% of the parameters is awesome! Crucial to model scaling.
2) Sweep-throughs in the ablations are comprehensive and help paint a clear picture.
3) Experimental results over a collection of datasets.
4) Open-source plans and I am sure this could be widely used and adapted

## Weaknesses
1) The experiments are limited to three smaller-scale wiki-based QA datasets. While these are widely used benchmarks (minus SQUAD, which I would say is a bit too dated now), including a broader range of datasets, such as MS MARCO and BEIR, would provide a more comprehensive validation of PSPT’s generalizability to ad-hoc search.
2) Missing baselines. I believe that this is missing a lot of baselines to situate it with other methods. Doc2query-T5 is an example of a simple model that should be included in my opinion, especially given this is similar to a query generation task. RankLLaMA/RankVicuna/RankZephyr too are two widely used baselines of roughly the same (RankLLaMA is the same) parameter count. Without such comparisons, the community can't situate the true value of the work.
3) I see Ret.Train is mentioned in the table but I believe it is never explained in the text. What filtering is this?
4) Missing exploration of how much train set/test set leakage exists.
5) I think the work misses a comparison to full fine-tuning. I believe the full-finetuning discussed is specific to the PSPT module. How much does full fine-tuning affect scores?

---

### Official Review · Reviewer_KCeZ · 2024-05-27
**This paper proposes a passage-specific prompt tuning method for reranking in open-domain question answering which adds a learnable embedding layer. Extensive experiments were conducted utilizing the Llama-2-chat-7B model on three datasets.**

**Rating:** 1
**Confidence:** 5

**Review:**

This paper proposes to use a parameter-efficient tuning approach for passage reranking in QA tasks.
A comprehensive set of experiments is conducted on three widely used open-domain QA datasets.
However, it is not clear why to use a passage-specific embedding layer. Can other parameter-efficient tuning methods like Lora, Adapter, and etc do the same thing? And there are also no comparison experiments with the SOTA reranking methods such as RankGPT.

---

### Decision · Program_Chairs · 2024-05-31

**Decision:**

Accept

**Comment:**

This paper proposes a method for learning soft prompts that are specific to a given passage to improve the ranking of passages. Reviewers noted that including more comprehensive experiments—such as stronger baselines, ablations for negative document selection, and comparisons with other training strategies—would be helpful. However, their overall comments suggest that this is an interesting approach supported by reasonable experiments.